# Enshrining Gender in Monuments to Settler Whiteness: South Africa's Voortrekker Monument and the United States' This Is the Place Monument

**Cynthia Prescott** [1,*], **Nathan Rees** [2] **and Rebecca Weaver-Hightower** [3,*]

1   Department of History, University of North Dakota, Grand Forks, ND 58202, USA
2   Department of Art, History, and Philosophy, University of West Georgia, Carrollton, GA 30117, USA; nrees@westga.edu
3   Department of English, Virginia Tech, Blacksburg, VA 24060, USA
*   Correspondence: cynthia.prescott@und.edu (C.P.); Rebeccawh@vt.edu (R.W.-H.)

**Abstract:** This essay examines two monuments: the Voortrekker Monument in South Africa and the American This is the Place Monument in Utah. Similar in terms of construction and historical purpose, both employ gender as an important tool to legitimize the settler society each commemorates. Each was part of a similar project of cultural recuperation in the 1930s−1940s that chose as their object of commemoration the overland migration in covered wagons of a group of white settlers that felt oppressed by other white settlers, and therefore sought a new homeland. In a precarious cultural moment, descendants of these two white settler societies—the Dutch Voortrekkers of South Africa and Euro-American Mormons (Latter-day Saints or LDS) of Utah—undertook massive commemoration projects to memorialize their ancestors' 1830s−1840s migrations into the interior, holding Afrikaners and Mormons up as the most worthy settler groups among each nation's white population. This essay will argue that a close reading of these monuments reveals how each white settler group employed gendered depictions that were inflected by class and race in their claims to be the true heart of their respective settler societies, despite perceiving themselves as oppressed minorities.

**Keywords:** monuments; the Voortrekker Monument; This is the Place Monument; Mormon; Latter day Saints; Afrikaner; Boer; settler; memorialization; whiteness





## 1. Introduction

In the early twentieth century, Western European nations and their settler societies around the globe celebrated military and political achievements by erecting elaborate public monuments. In South Africa and the western United States, the focus of this essay, these monuments celebrated the planting of new settler societies and—at least implicitly—the conquest of indigenous lands and peoples (Prescott 2019). These public commemorations of European-descended explorers, statemen, and generic mother figures were grounded in idealized depictions of individuals as paragons of particular raced and gendered cultural values, presenting implicit arguments for the legitimacy of settler society. As such, it is not surprising that these same monuments in more recent years have served as sites of resistance for a changing populous that has sought to reframe the cultural conversation. For instance, South Africans and Americans in recent years have targeted some monuments for removal, most famously in the "Rhodes must fall" movement in Cape Town and in efforts across the US to remove monuments to the American Confederacy. This essay examines two monuments not from those movements that, though part of the same larger conversation about material culture and memory, have not drawn the same level of controversy as the Rhodes and Confederate monuments: the Voortrekker Monument in South Africa and the American This is the Place Monument in Utah. Similar in terms of construction and historical purpose, both also employ tropes of gender to justify the settler society each

commemorates. As this essay will demonstrate, settler societies employed representations of gender, alongside race, as an important tool in legitimizing colonization.

The remarkably similar historical background of each monument is what brings them into comparison. Each was part of a project of cultural recuperation in the 1930s−1940s that chose as their object of commemoration the overland migration in covered wagons of a group of white settlers that felt oppressed by other white settlers, and therefore sought a new homeland. In a precarious cultural moment, descendants of these two white settler societies—the Dutch (or "Boer") Voortrekkers of South Africa and Euro-American Mormons (Latter-day Saints or LDS) of Utah—undertook massive commemoration projects to memorialize their ancestors' 1830s−1840s migrations into the interior, holding Boer-descended Afrikaners and Mormons up as the most worthy settler groups among each nation's white population (Prescott et al. 2020) To do so, both draw on a European tradition of building massive granite or granite-faced structures with relief sculptures on the interior or exterior (see Figures 1 and 2). In South Africa, architect Gerard Moerdyk and a team of four sculptors designed a three-story building adorned with sculpture to commemorate the Voortrek and its leaders to emphasize Afrikaner heroism and cultural dominance in the wake of a disastrous civil war loss to the British. Their commemoration of the Boer Voortrekkers' attempted settlement on indigenous-African-controlled lands supported politically ascendant Afrikaners, who would soon introduce racist apartheid policies. In the US, artist Mahonri Young, grandson of LDS prophet and Utah territorial governor Brigham Young, created the This is the Place Monument to celebrate the centennial of the Mormons' establishment of their settlement in Utah and the five-decade work of integrating into the US culturally as well as legally (Patterson 2020, pp. 31–55). An important part of the project meant depicting the Mormon minority as "all American" while erasing the taint of polygamy.[1]

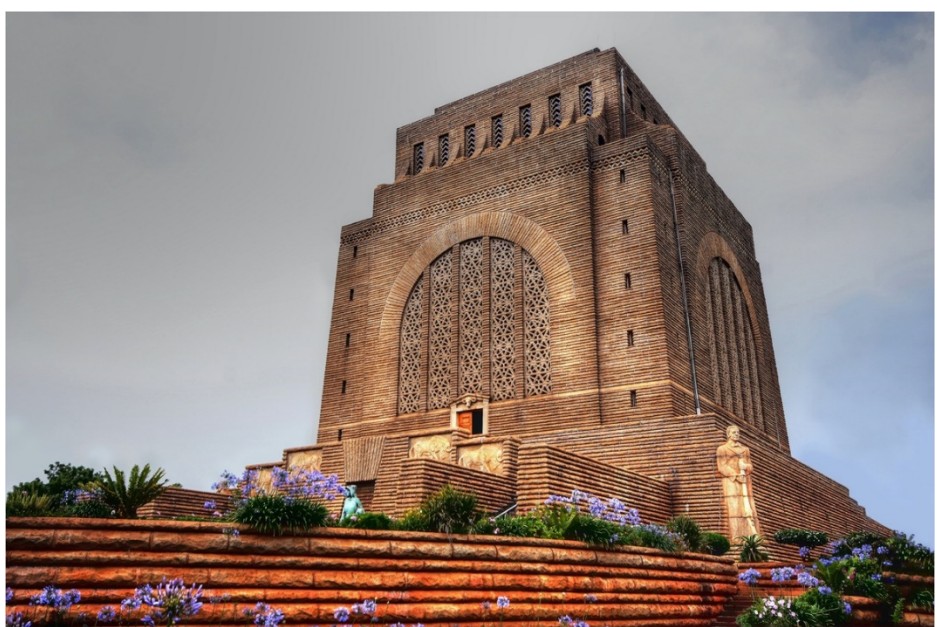

**Figure 1.** Voortrekker Monument exterior. Source: Wikimedia Commons, https://commons.wikimedia.org/wiki/File:Voortrekker_Monument_922580097.jpg (accessed on 25 February 2021).

---

[1] Beginning in secret in the 1830s, and then openly after 1852, Latter-day Saints practiced a form of polygyny they termed "plural marriage," wherein leading Mormon men married multiple wives simultaneously. The LDS Church ended the practice beginning in 1890. See (Ulrich 2017).

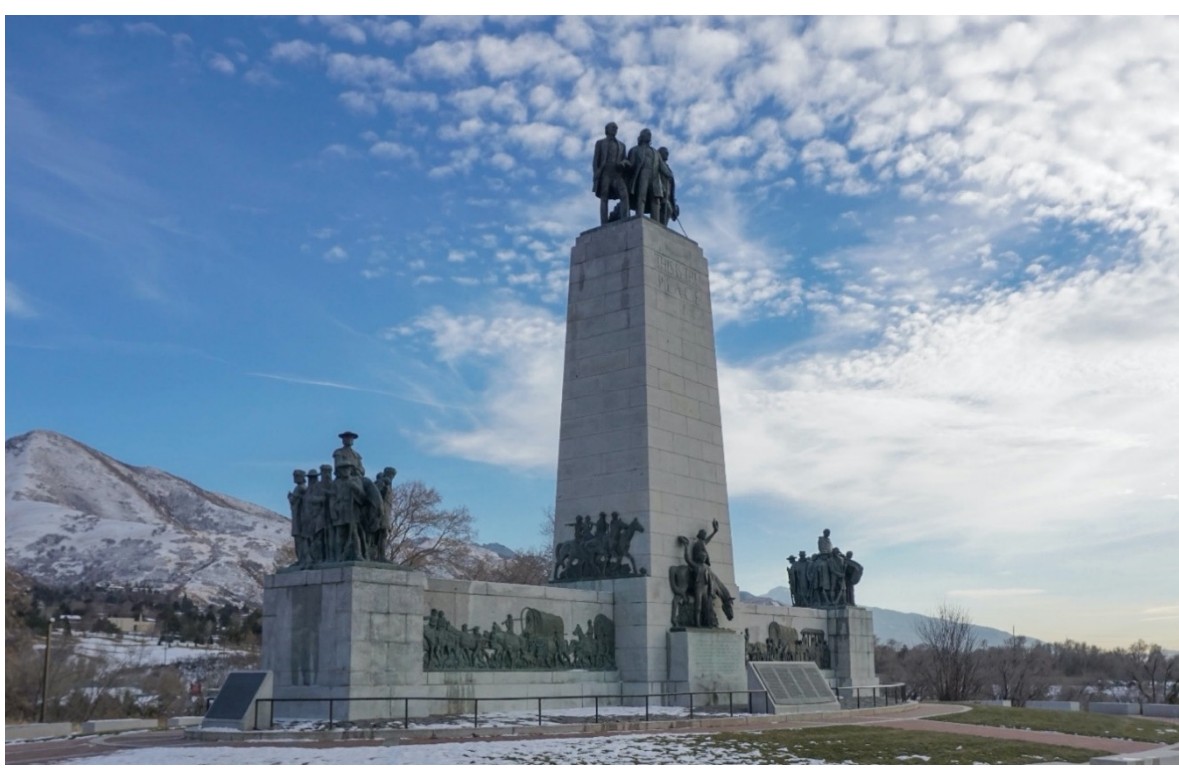

**Figure 2.** This is the Place Monument. Photograph by Nathan Rees.

This essay will argue that a close reading of these monuments reveals how each white settler group employed gendered depictions that were inflected by class and race in their claims to be the true heart of their respective settler societies, despite perceiving themselves as oppressed minorities. That is, both monuments to the achievements of great white men also contain culturally significant depictions of lesser white men, indigenous men, and white women, but erase indigenous women, all of which this essay will analyze. Moreover, while examining similarities in depictions of hegemonic white masculinity and white motherhood, our comparison brings to light important differences in the depiction of settler women's roles that would not be as apparent if reading either monument alone. In order to make this argument, this essay first examines depictions of masculinity, specifically how comparison reveals the extent to which the Voortrekker and This is the Place monuments emphasize racial and religious conquest through specific masculine achievements. Next the essay turns its eye to the monuments' depictions of settler women, especially sunbonneted white mothers carrying Western "civilization" into "savage" interior lands. The Voortrekker Monument represents the South African nation in the body of the white "Volksmoeder" ("mother of the nation"), excluding indigenous women—many of whom raised white children as domestic servants—from its ideal. Utah's This is the Place Monument, by contrast, erases Mormon women's contributions to settlement, obscuring the settlers' polygamous social order in order to rehabilitate Latter-day Saints' image. Comparing these monuments' racialized constructions of gender exposes the ideological work of these material objects while offering a framework for understanding the role of gender in monuments to colonialism.

## 2. Constructing Settler Masculinity

Examining the monuments' representations of masculinity shows how both create a version of history where dominant Boer or Mormon men—with the support of other white men—*naturally* dominate women and indigenous men. Both Afrikaners and Mormons perceived themselves as oppressed, and the men of each group sought to avoid being dominated by other white settler men. As such, both monuments employ idealized masculinity

to celebrate European-descended groups that withdrew from "white" settler society in the 1830s and 1840s and moved to supposedly "unsettled" interiors—the Boer Voortrekkers moving out of the Cape region into the interior of southern Africa, and the Mormons from the Midwest into the interior southwestern Great Basin of North America.[2] By the 1930s, the Afrikaners and Mormons were gaining influence in their respective nations, and both turned to elaborate monumentalization to rehabilitate their image and enshrine their supposed cultural superiority not only to the indigenous populations they had violently displaced, but also to the other white settler groups. In the late nineteenth century, Dutch South Africans and Mormons had both suffered significant political crises: Afrikaners losing to British settlers in the second Anglo-Boer War, and the Mormons sacrificing their vision of an independent, polygamous theocracy to enable US statehood. So, in their twentieth century memorializations, each group created explicitly racialized and gendered tributes to their earlier nineteenth-century emigrant forebearers, emphasizing the achievements of male leaders while presenting indigenous men as either hyper-masculinized or feminized, in both cases out of line with the supposed natural order of settler culture and civilized manly behavior.

As recent theorists have argued, gender expectations and norms provided such powerful tools of imperial ideology because, throughout Western history and culture, gender has been considered "natural," representing the divinely ordained order of the universe, with white men being naturally ascendant over white women, children, and all non-white peoples. Critics including Irving Goffman and Judith Butler have long argued that gender, as a social construction, is not natural; and critics like RW Connell and Anne McClintock have argued that gender, class, and race are co-determinate and hegemonic (Goffman 1979; Butler 1990; Connell 1995; McClintock 1995). But in the mid-nineteenth century, the conception of a natural order meant that high-status white men in the US and South Africa pursued a sense of refined manliness demonstrated by gentility in tastes and manners and defined against a lower status masculinity (exhibited by most settlers) grounded in work and labor. But by the early twentieth century, similar gentlemen fearing they had become *too* civilized (i.e., feminized) led to organizations such as the Boy Scouts that offered opportunities for urban youth to mimic the supposed rugged masculinity of mid-nineteenth-century settlers or assume the indigene's perceived physical prowess while maintaining the bounds of whiteness (Kimmel 2006; Bederman 1996).

Sculptors of the Voortrekker and This is the Place monuments, we will show, merged these competing notions of refined manliness and rugged masculinity to portray white settlers overall as masculine but civilized. However, both monuments use clothing, posture, and sculptural positioning to portray high-status Boer and LDS leaders as equal or superior to competing male settlers in their respective nations, and to differentiate leaders from supporting men of a lesser class, indigenous men, and—as we will later discuss—women. The monuments depicted other Boer and LDS settlers as supporting these leaders, yet established these men's masculinity through labor and displays of civilized dominance over white women and indigenous men. In sharp contrast, the monuments portrayed indigenous men as either hypermasculine or feminized to assert that they were unworthy of land ownership, thus securing settler claims to those lands through their properly masculine and "civilized" leaders. Comparing the monuments' gendered constructions of leaders, other white men, and indigenous men demonstrate how Afrikaners and Mormons employed gender as a tool to justify similar systems of oppression and privilege.

In South Africa, the creators of the Voortrekker Monument drew upon the perceived masculine power of early nineteenth-century Boer settlers to assert twentieth century political dominance over indigenous people and other white settler populations. Divided by Dutch and British ancestry, South African settlers staked out competing colonial outposts in the area that was to become Cape Town: the Dutch in 1652 and the British in the early

---

[2]  By "white," we refer to settlers with primarily European ancestry. We follow critical race studies' project of unpacking the ways in which European-descended settlers constructed and employed concepts of whiteness and blackness. See Ingram (2001), "Racializing Babylon," and Reeve (2015), *Religion of a Different Color* on the racial construction of whiteness by each settler society.

nineteenth century as part of spoils from the Napoleonic wars. The two groups competed for control of South Africa, including two civil wars in the late nineteenth and early twentieth centuries. After the Afrikaners lost the second Anglo-Boer war, they sought to recuperate their self-image into one of heroism and success, in part by remembering an incident of earlier-nineteenth-century warfare where one of the trek Boer groups fought and defeated the Zulu in battle, a victory that Afrikaners viewed as God's sanctioning of white settler rule in that space. The Voortrekker Monument served as a monumental shrine to that event and its white male leaders, helping to make it a rallying cry for Afrikaner nationalism.

The Voortrekker Monument emphasized these Voortrek leaders by featuring heroic-sized, physically elevated sculptures at each of the structure's four corners. Visitors reach the monument by ascending hundreds of stairs, so that, as Figures 1 and 3 illustrate, these slightly elevated figures are even more imposing upon approach. The granite sculptures depict Boer leaders Piet Retief, Andries Pretorius, Hendrik Potgeiter and a generic male figure representing the remaining Voortrekkers. All four gatekeepers or "guards of honor" wear the typical Voortrekker costume of short buttoned jacket and baggy, ankle-length trousers (Rankin and Schneider 2020a, pp. 310–11), and each holds a rifle in front, implying that the hegemonic white men who guard the building also guard the culture (see Figure 3). They are called out as dominant over and above the women, non-white people and even other men incorporated into the monument's interior, as well as the British men not depicted on the exterior of the monument but who these patriarchs overlook in Pretoria below.

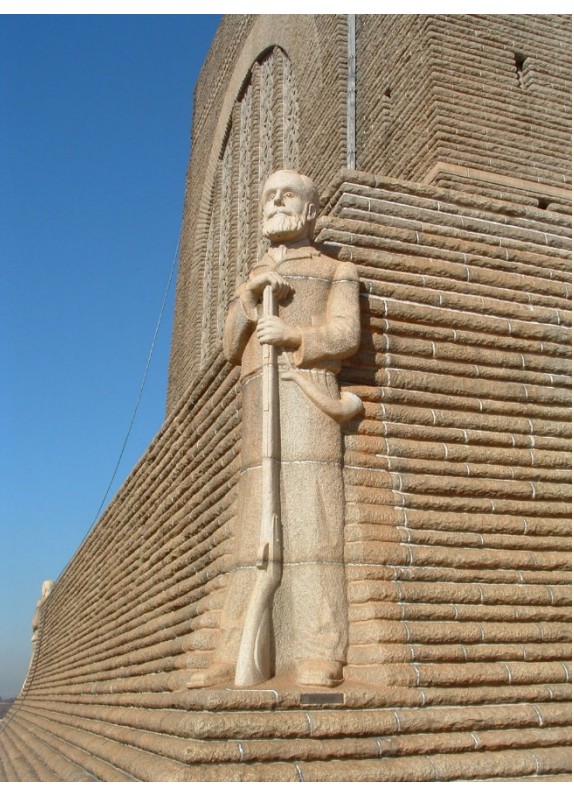

**Figure 3.** Piet Retief as a member of the "Guard of Honor" on the exterior corner of the Voortrekker Monument. Source: Wikimedia Commons, https://commons.wikimedia.org/wiki/File:Voortrekker_Statue.JPG (accessed on 25 February 2021).

The sculptural frieze on the interior of the building likewise celebrates achievements of white, male Boer leaders, by presenting a story of the great treks combined into one linear narrative. The 92 m (302 foot) long white marble frieze encircles the entirely of the massive

interior space of the Hall of Heroes.[3] It features an Exodus-styled journey, beginning with a panel entitled "Voortrekkers Leaving the Cape Colony, 1835," through 26 other bas-relief panels presenting bonneted stern-faced women and children and bearded men, first in discussion with and then in conflict with hordes of nearly naked male indigenous people. Though the unpainted marble makes all of the figures literally white, it is clear from the clothing and phenotypes of the figures who is "white" (Afrikaner) and who is "black" (indigenous) and with whom visitors are meant to identify.

All Afrikaner men depicted in the frieze wear European-style clothing, including coats, collared shirts, and trousers. But artist H.W. Coetzer (who began the design work on the Voortrekker frieze) and sculptors Peter Kirchhoff, Frikkie Kruger, Laurika Postma and Hennie Potgieter (who executed it) differentiate Boer men both from over-refined European men and barbaric African foes through positioning and distinctive attire.[4] While British and Portuguese soldiers wear lavishly ornate decorations on their uniforms that emphasize their power to the point of feminizing them, heroic Boer leaders match the civilian British leaders' formal attire of long coats, bow ties, and top hats, and thus their high status without the pomp.

The frieze also uses clothing and position to place other male trekkers (wearing short, buttoned coats and wide-brimmed hats) above nearly naked indigenes. When doing the hard labor required to dominate the landscape in the absence of other ethnic groups, the frieze depicts some ordinary trekkers as masculine laborers working in their shirt sleeves. But when Boer men appear in the presence of the dominant British or scantily-clad indigenes—depicted with bare chests and legs and only an apron tied about their waist—the frieze shows Boer leaders in formal attire and all Boer men wearing buttoned short coats, even in the heat of battle, lest their civility be questioned. Moreover, the frieze regularly positions Boers as separate from and above their low-status black foes (see Figure 4). In fact, the appearance of a slain Boer man in "Murder of Retief," with his jacket and collared shirt torn aside, revealing the indignity of his bare torso, emphasizes the barbarity of African attackers and helps cast Retief and his men as martyrs to a noble cause. This juxtaposition of Boer leaders against lesser Boer men and indigenous figures emphasizes the hegemonic masculine authority of the leaders, while still demonstrating the masculinity of even the lower-status Boer men.

Although in reality the Voortrekkers invaded the lands of the Ndebele and Zulu, six of the twenty-seven frieze panels depict hordes of generic, stereotyped, anonymous and half-naked indigenous African men armed with clubs or spears savagely attacking formally attired Boer settlers sculpted with distinct facial features granting them individuality. The Boers' painstakingly historically accurate material culture—formal clothing, wagons, wooden furniture, leather and glass goods, and firearms—emphasize the veracity of this imagery and their cultural dominance over "primitive" indigenes armed with spears. As Figure 4 shows, African leaders stand out from their followers with ornate jewelry and headpieces. That they still bare their chests, arms, and legs marks them as less civilized than African servants who wear a loose-fitting woven shirt or trousers to serve their Boer masters, both clearly inferior to even the most humble Boer folk. Moreover, historians have uncovered the great pains taken in the frieze planning and construction to emphasize this distinction, even, as shown in Figure 4, making sure that the indigenes are physically below the white settlers in their positioning in the frieze (Rankin and Schneider 2020b, p. 265. And as Figure 5 shows, the frieze portrays the Zulu warriors as disordered, out of control, and animalistic (one even prostate on the ground) in contrast to the rows of equestrian Afrikaner soldiers, who despite being outnumbered, manage to ride on horseback while shooting horizontally-aligned guns, their coats buttoned and orderly, hats neatly upon

---

[3]  Though the frieze panels themselves have minimal captioning, they have long been explained by an eighty-four-page supplementary text offering an extensive history of the monument's construction, symbolism, and ideology, *The Voortrekker Monument Pretoria, Official Guide* (Board of Control of the Voortrekker Monument 1955).

[4]  Coetzer originally intended the Voortrekkers to appear as ordinary folk, with shirt sleeves rolled up for work. But the monument-planning historical committee insisted on propriety, such as that Boer men wear buttoned jackets. Rankin and Schneider (2020a, p. 169), *From Memory to Marble*.

their heads. These images draw from tropes of disordered violent hypermasculinity to create a threatening Other as outside of the bounds of civilized masculinity both in the 1830s and 1930s, and in both times not to be trusted with land ownership. Moreover, the image of trek Boers as properly masculine men dominant in battle works to recreate an image of Afrikaner superiority threatened by the loss of the second Boer war.

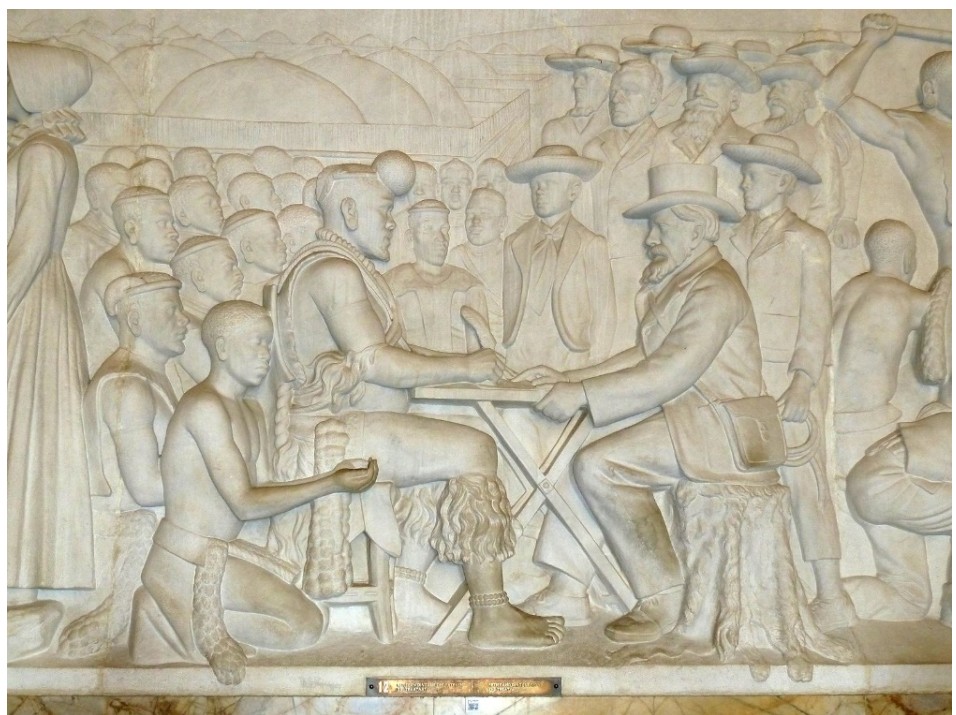

**Figure 4.** Panel 12 from the Voortrekker Monument frieze, showing Retief and Dingane signing the treaty which grants the Voortrekkers land to the south. Source: Wikimedia Commons, https://commons.wikimedia.org/wiki/File:Dingaan_en_Retief,_paneel_12,_Voortrekkermonument.jpg (accessed on 25 February 2021).

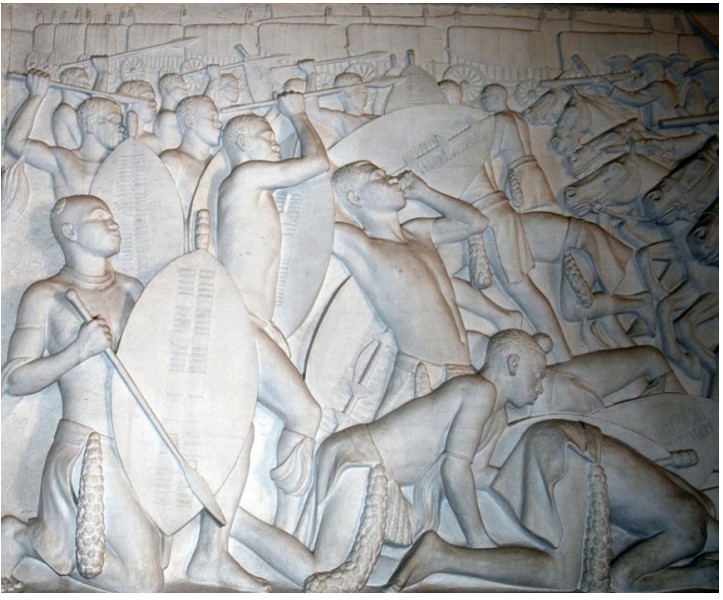

**Figure 5.** Detail from panel 21, "Battle of Blood River." Source: Public Domain Pictures, https://www.publicdomainpictures.net/en/view-image.php?image=89858&picture=the-battle-of-blood-river (accessed on 25 February 2021).

With minimal captioning, visitors not familiar with the history of the treks (or the years of planning, debating and creating the frieze) would see a narrative of inevitable and righteous conflict, which the Boers tried to avoid through treaty until betrayed and forced into a battle that they won through divine aid. Interestingly, the frieze does not end with triumph over the indigene but instead with the Voortrekkers leaving the region "after the British occupation" and the British recognizing the independence of the Transvaal. The narrative becomes, then, Afrikaner distinction from and triumph over the Zulu (and by extension all black South Africans) as well as the British. The narrative emphasizes triumph over adversity, indigeneity, and colonial competition led by civilized Boer male leaders with the support of lesser, yet still masculine Afrikaner males—and, as we will show, properly feminine female—settlers.

Tropes of masculinity are equally present in This is the Place, which, like the Voortrekker Monument, was erected to commemorate a past victory in the service of refashioning its settler group's public image in the mid twentieth century. Founded by prophet Joseph Smith in 1830 in upstate New York, the LDS Church's followers repeatedly fled persecution by migrating westward. After Smith was killed in mob violence, his successor, Brigham Young, led a majority of Mormons to the Great Salt Lake in a region that was already inhabited by the Northern Shoshone and Timpanogos Ute indigenous groups (Christy 1978). The monument's title refers to Brigham Young's alleged declaration at his first view of the valley—from the site where the monument was placed.[5]

More than merely separating themselves from US society, nineteenth-century Mormons attempted to build a polygamous theocracy in the Great Basin—a project that met wide resistance from other white populations in the United States who cast Latter-day Saints not only as un-American, but as less than fully "white," associating the practice of polygamy with a savage racial Other (Reeve 2015). By the 1890s, Latter-day Saints were forced to abandon polygamy and yield much of their political and civic control for Utah to achieve US statehood. Decades later, twentieth-century Mormons working to gain acceptance within American society downplayed their former radical marriage practices and cast their mid-nineteenth-century migration as part of—or even the epitome of—the US overlanding experience by erecting pioneer monuments to celebrate prominent Mormon men.[6]

Both monuments invoke the perceived natural hierarchy of gender to carry out their project of cultural recuperation. Even more so than the Voortrekker Monument, This is the Place focuses on prominent male leaders. Sculptor Mahonri Young placed his grandfather, prophet Brigham Young, in a central position, standing atop the central pillar, accompanied by LDS leaders Heber C. Kimball and Wilford Woodruff (see Figure 6). The three men's position of height relative to the viewer and other monument figures emphasizes their command over the space and its people, both white and indigenous.[7] Other well-known LDS men appear around the base of the central pillar, but their placement around the pillar and on horseback emphasizes their dominance over other, lesser men elsewhere on the monument (see Figure 2). Below them, the first company of LDS settlers who accompanied Brigham Young to Utah drives wagons across the front of the monument's long base. Most are mounted on horseback or carry rifles like those in the Voortrekker Monument to demonstrate their masculine power. Prominent non-LDS westerners stand in groups at the corners of the base and line the back of the monument like the Voortrekker Monument's external sentries. Their placement below LDS leaders and behind the First Company, combined with their less refined attire, indicate their lower social status and suggest that these overly or insufficiently masculine men paved the way for the Mormon migration and founding of Utah.

---

[5]   On the contested historicity of this event, see (Poulsen 1977, pp. 246–52).

[6]   On Latter-day Saints' assimilation efforts, see (Mauss 1994; Alexander 1996).

[7]   (Goffman 1979). As Goffman recognized, part of establishing this masculine hegemony has to do with their relative size and function ranking.

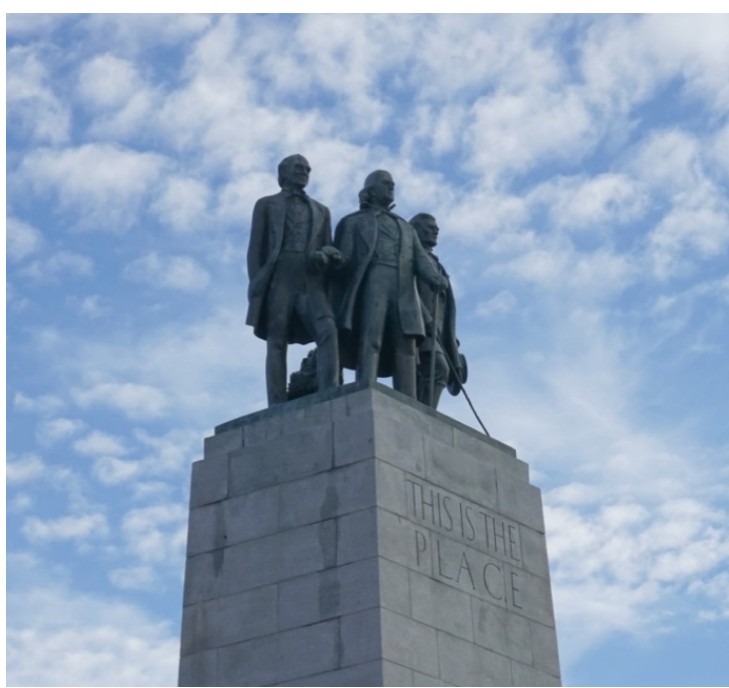

**Figure 6.** Mahonri Young, Heber C. Kimball, Brigham Young, and Wilford Woodruff atop the central plinth of This is the Place Monument, 1947. Photograph by Nathan Rees.

The attire of these various figures in This is the Place establishes their racial and class status, reinforcing the message of the inevitability of LDS dominance and their status within the hierarchy of hegemonic masculinity. At the monument's top, the highest-status Mormon men wear formal long coats, waistcoats, and stock ties, establishing their patriarchal authority through their class status. The LDS men placed below Young, Kimball, and Woodruff are less formally attired than the Mormon leaders, thus supporting the authority of the three leading figures on the highest plinth. Like Boers in the Voortrekker Monument frieze, laboring LDS men encircling the central plinth and driving the wagon train across the front of the monument's base wear collared shirts, sometimes topped with jackets or vests. The most informally attired ride horses or carry rifles to reinforce their status as civilized men above other white men who failed to permanently settle the Great Basin and above the region's indigenes.

Atop the two ends of the wide base are groups of non-Mormon white men wearing somewhat exotic attire. Like the Voortrekker frieze British and Portuguese soldiers, the group of Spanish missionaries in long robes accompanied by soldiers in sweeping capes appear effeminate by 1940s standards, despite having guns and a horse. The Anglo-American fur traders on the other corner and the mountain men on the monument reverse wear fringed coats and rustic animal skin caps that link them to overly aggressive, animalistic indigenes. The non-Mormon's less civilized attire and implied engagement with indigenous people place them lower on the social hierarchy than the Mormon leaders on the central plinth and even the lower-status LDS men on the monument's base who used their knowledge of the region to settle. In a large plaque on the monument's reverse, the men of the doomed non-Mormon Donner Party—who attempted to carve a wagon road through Emigration Canyon in 1846, literally clearing a path through which Young would lead the LDS First Company the following year—strain like animals against ropes to move a single wagon. Not only do they remove their jackets and roll up their shirt sleeves like laboring Voortrekkers and the men of the LDS First Company on the front of the monument's base, but three of these non-LDS men even labor naked from the waist up. Yet like the shirtless black servant in the Voortrekker frieze (but unlike that frieze's even less clothed indigenous warriors), these white men's hypermasculinity is constrained: they turn their backs to the viewer rather than reveal their nakedness.

Like most American pioneer monuments of the era, This is the Place largely erases indigenous peoples from its narrative of white settlement, implying that the Great Basin was a *terra nullius* awaiting LDS settlement (Prescott 2019, chaps. 1–2). The only indigenous figure included in the monument is Washakie (Figure 7), a male Northern Shoshone leader who temporarily joined the LDS Church and allied with Mormon settlers (Patterson 2020, p. 45). By presenting Washakie as the only indigenous individual in the narrative of Utah settlement, and by placing him on the back, surrounded by and in the shadow of Spanish missionaries who preceded LDS attempts to convert the region's indigenous populations, This is the Place indicates that the only acceptable role for Native Americans in Mormon Utah was collaboration. Excluding images of Native warriors like those that appear in nearly half of the Voortrekker frieze panels reinforces Mormon fantasies of Utah as an empty land to which the LDS were called by God and evokes the popular American trope of the "vanishing Indian." (Blee and O'Brien 2019; Brantlinger 2003) Excluding Native women further ensured that indigenous populations would disappear, yielding space to the descendants of LDS pioneer mothers. While the indigenous Other in South Africa was always numerically dominant and therefore unerasable, This is the Place reassures Mormons and all Americans that their taking of indigenous lands was justified, both by a fantasized lack of indigenes, and because settlers were welcomed by the indigenes who were present, like Washakie.

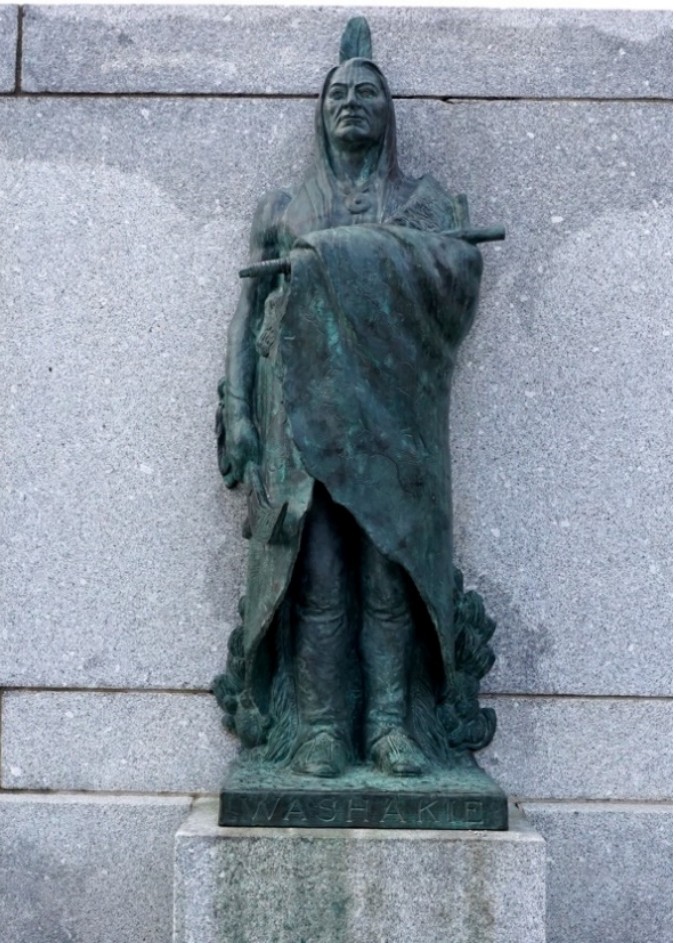

**Figure 7.** Mahonri Young, "Washakie," from This is the Place Monument, 1947. Photograph by Nathan Rees.

Moreover, the depiction of Washakie feminizes him. In contrast to extant photographs of the Native leader wearing Western dress, Mahonri Young chose to portray Washakie wearing an indigenous robe draped over his otherwise bare chest. Washakie's long hair,

soft facial features, and slender build and the peace pipe he cradles—signaling his intention to yield to LDS civilization—all place him in the position of the feminized Other, following contemporaneous representations of "pacified" Native Americans as effeminate (Scott 2011, pp. 20–47). Of course, by excluding Native American women from this and other US pioneer monuments altogether, they hide the reality of miscegenation on the frontier and further the fantasy that the region will only produce civilized white society (Prescott 2019, chap. 1).

Both This is the Place and the Voortrekker Monument use gendered imagery to create a racialized Other outside the bounds of acceptable masculinity, whether not masculine enough or too masculine, juxtaposed with the white settler (and especially the settler leader) as the exemplar of civilized masculinity. While the Voortrekker frieze portrays hypermasculine black Africans as animalistic and subhuman, This is the Place presents a feminized Native American as more assimilable; those who fail to convert to Mormonism or other lesser Christian faiths simply disappear, yielding to white men's dominance. And because both Boer and Mormon men appear as strong and authoritative, civilized but not effete, their settlement projects are implied to be even more just and more divinely ordained than those of other European-descended groups with whom they competed for land in the 1830s and with whom they jockeyed for equality (or superiority) in the 1940s.

## 3. Containing Women

Just as This is the Place and Voortrekker monuments employed similar portrayals of 1840s hegemonic white masculinity to support the cultural project of the 1940s, they likewise utilized racialized and class-specific imagery of white femininity. Both 1940s monuments looked back to similar ideals for womanhood that had emerged in western Europe and in European settler societies in North America and South Africa in the mid-nineteenth century. However, in their parallel efforts, they each diverged in important ways in representing idealized nineteenth-century white womanhood. The Voortrekker Monument embraced a self-sacrificing Volksmoeder that paralleled the sainted US Pioneer Mother. This is the Place, in contrast, downplayed pioneer mothers and demoted Mormon women from iconic embodiments of white civilization to obedient supporting figures, obscuring the realities of LDS women's exceptional contributions to nineteenth-century Utah to avoid the lingering taint of polygamy.

Just as with masculinity, however, femininity is hegemonic, meaning that women have been and are defined in relation to men, children, and other women according to class and race. In the nineteenth century, western Europe and its settler societies embraced an emphasis on women's domestic labor and reproduction. Higher status women typically set themselves apart by focusing on decorative tasks such as needlework, while quotidian tasks such as laundry—let alone physically demanding field labor—were erased or passed off to servants of a lower class and/or racialized status (Brink 1990, pp. 273–92; Petzold 2007, pp. 115–31; Prescott 2007, chap. 1). Despite the realities of physically demanding labor, settlers struggled to promote that domestic ideal. So, for instance, to protect their pale skin, juxtaposed against darkness and non-European physiognomy, female settlers in the US, Canada, and South Africa wore wide-brimmed bonnets. That those same sunbonnets (or kappies, as they were known to the Dutch-speaking Boers) also constrained women's vision and actions was no accident, and in public memory over the course of the twentieth century in the US and South Africa, those sunbonnets and kappies came to symbolize the very cultural ideals of longsuffering nurturance to which the women who wore them were held in the mid-nineteenth century (Matheson 2009; De Beer 2009). In the early twentieth century, cultural memory of those settler women converged into a popular image of a white mother whose suffering paid for twentieth-century settler societies' relative ease, prosperity, and uncontested land ownership (Prescott 2019, chaps. 2). Clad in a long dress and sunbonnet/kappie and accompanied by children, the South African Volksmoeder and American Pioneer Mother were symbolic mothers to their respec-

tive nations, their purity embodying white dominance in each settler society (Brink 1990; Petzold 2007; Prescott 2019, chap. 2).

Like many US pioneer monuments of the same era, the Voortrekker Monument places a bronze Volksmoeder in a position of honor at the monument's entrance. As Figure 8 shows, she stands gazing out past the viewer, her white face protected by a wide-brimmed kappie, while the children tugging at her long skirts look to her for support and comfort (Prescott 2019, chap. 2, cf). Sculpted by Anton von Wouw—known for the 1913 Vrouemonument depicting Boer women suffering in British concentration camps during the Boer wars, which represented one of the earliest visual representations of Volksmoeder ideology (Marschall 2010, pp. 238–39)—she stands at eye level at the center of the north façade, placed in comparison with but physically below the four guardian patriarchs at the building's corners, as was believed to be her natural station. Like those simply-dressed male trekkers, her kappie and plain gown establishes her middling class status. The Volksmoeder is defined through her relationships to the Boer men above her, the children at her feet, and four wildebeests to either side of her. Standing in for the indigene and symbolizing the triumph of civilization over savagery, these wild creatures ensure her purity by allowing the avoidance of black African men assumed to be animalistic and prone to sexual violence against women. The *Official Guide* to the monument says that the wildebeests "portray the ever threatening dangers of Africa," though their "retreating attitude . . . suggest that the dangers are receding and that the victory of civilization is an accomplished fact" (Board of Control of the Voortrekker Monument 1955, p. 35) Black women, as we will discuss, are completely erased.

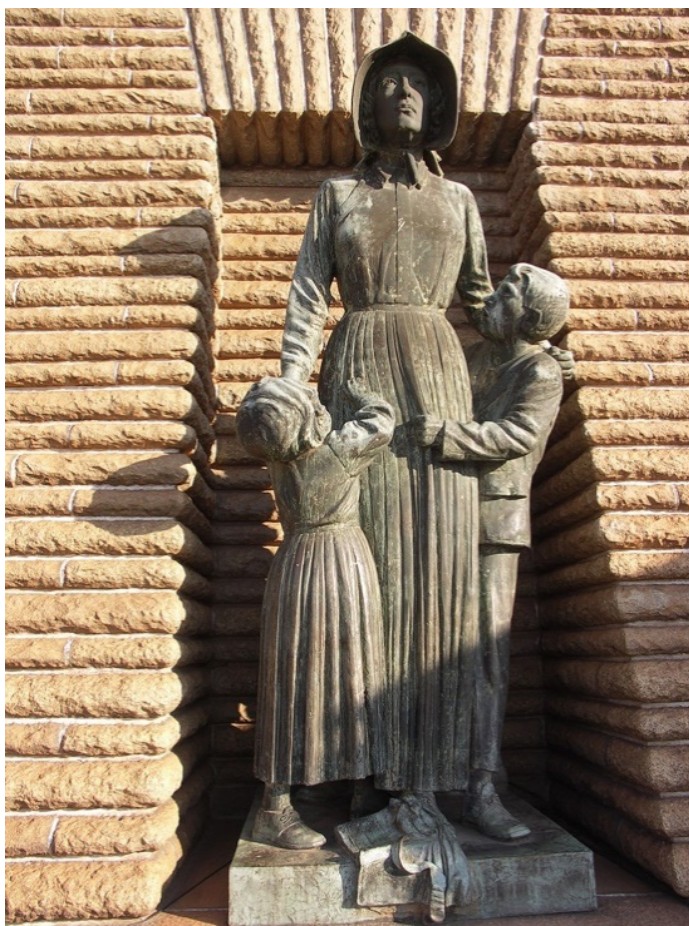

**Figure 8.** The Volksmoeder at the entrance to the Voortrekker Monument. Source: Wikimedia Commons, https://commons.wikimedia.org/wiki/File:South_Africa-Voortrekker_Monument-Woman_and_Children01.jpg (accessed on 25 February 2021).

The frieze inside the Voortrekker Monument also contains multiple images of bonneted white women acting as mothers and supporting white male leaders. Visitors see images of settler women riding on wagon boxes or looking on as men navigate difficult wagon trails, women accompanying their husbands on the trek, and women doing domestic labor. But, additionally, the Voortrekker frieze shows Boer women playing active roles that expand the Volksmoeder ideal, which in the 1940s would help replace the image of the female concentration camp victim with the worthy helpmate. For example, the panel entitled "Women spur men on" (Panel 18) represents the popular notion that despite many losses, Boer women refused to admit defeat and leave Natal. Instead, the women demanded that the Boer men avenge settlers killed in conflict with indigenes, bearing up under difficulty when the men cannot do so on their own, supporting their men, as would be expected in the natural hierarchy of gender and family. Yet unlike oft-told stories of Susanna Smit's declaration that the women would prefer to walk barefoot over the Drakensberg Mountains than submit to British authority, this scene depicts unnamed women who gently encourage their despondent husbands.

Moreover, two other frieze scenes depict Boer women taking up male roles. "The women at *Saailaer*" (panel 23) depicts women doing agricultural and defensive labor while their men are away at war with African indigenes. "The Battle of *Vegkop*" even depicts Boer women taking part in the heat of battle. One woman even joins the men on scaffolding attached to the wagons to shoot the attacking Ndebele warriors.[8] Aside from a few women appearing without the protection of kappies in indoor or urban scenes and—more strikingly—while defending themselves against savage indigenes, the sculptors depicted all female trekkers as properly attired ladies. Afrikaners thus depicted Voortrekker women as active to help diffuse the memory of female victimhood at the hands of the British and remind of the "natural" order, where white women held authority over black men and women while helping their white men.

Despite including scores of black South African males, the frieze nearly excludes black South African women. Black South African women were, as Anne McClintock reminds, culturally significant in white settler society in the twentieth century, since they acted as domestics, nannies, and mother figures for white children of a range of classes (McClintock 1995, conclusion). But according to Rankin and Schneider, the only proposed depiction of a black servant woman quickly disappeared from early plans for the Voortrekker frieze (Rankin and Schneider 2020b, pp. 13–15). In fact, black African women only appear in one out of the twenty-seven frieze panels, and then as four of the many polygamous wives of a savage Zulu chief (panel 25, "Death of Dingane"). These African women wear only neckpieces, girdles, and small front aprons, in sharp contrast to the Boer women who are consistently depicted wearing long dresses and wide-brimmed kappies. By defining South African motherhood so firmly as white, Afrikaners of the 1940s disavowed the women who were intimate in their homes while celebrating a certain version of Boer femininity as part of their project of legitimizing settler rule.

Both the Voortrekker and This is the Place monuments depict self-sacrificing white mothers dressed in long gowns and wide-brimmed bonnets while erasing indigenous women. But juxtaposing their positioning relative to other characters in each monument's sculptural elements reveals a surprising disjuncture in the two settler societies' use of gender as a means of supporting their colonial legacy and proclaiming their contemporary superiority. While the Voortrekker Monument celebrates the heroic white Volksmoeder and the white Afrikaner family unit, This is the Place nearly erases the contribution of female Mormon settlers as part of sidestepping the issue of the polygamous Mormon family. This comparison between the monuments brings into relief how both used a raced and classed gender to reimagine their nineteenth-century settler forebears in the service

---

8  (Rankin and Schneider 2020b, pp. 79–99). *From Memory to Marble* contains excellent reproductions of all of the frieze panels, including those referred to in this section.

of twentieth-century efforts to establish themselves as integral to the narrative of national expansion.

This is the Place differs from most early-twentieth-century American pioneer monuments, which celebrated a self-sacrificing Pioneer Mother that bears striking resemblance to the Voortrekker's Volksmoeder (Prescott 2019, chap. 2). In its representation of a non-Mormon woman—the female member of the Donner Party represented on the monument's reverse in Figure 9—This is the Place offers a particularly servile Pioneer Mother; whereas the Voortrekker Volksmoeder and best-known US Pioneer Mother statues are placed centrally facing the viewer, the Donner Party figure trails behind her wagon company. We see the non-LDS Pioneer Mother almost in profile, turned outward toward the viewer just enough to reveal her face. While the men around her move with decisive, physical action, straining against the difficult mountain terrain, the Donner Party mother trudges along, carrying a baby and leading her young son. Despite the importance of her role as a pioneer mother, she is clearly subordinate to the men leading her company.

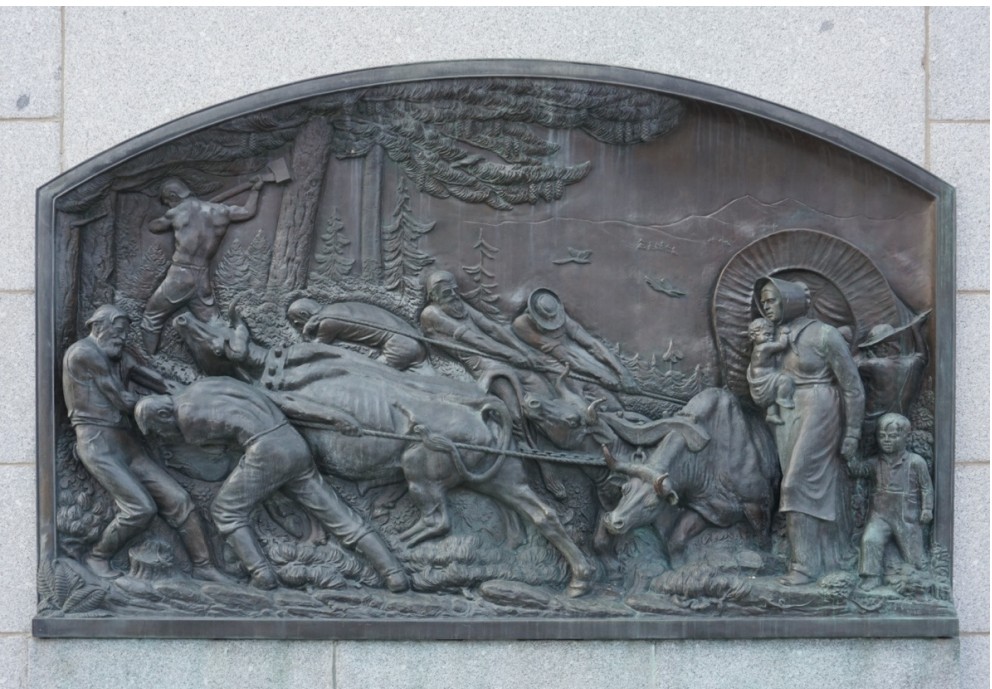

**Figure 9.** Mahonri Young, Donner Party from This is The Place Monument, 1947. Photograph by Nathan Rees.

Rather than representing Mormon women in more powerful roles, This is the Place was even *more* circumspect of their contributions. Unlike the Voortrekker Monument's celebration of Boer women's assistance to their men, This is the Place represents only three nearly-anonymized Mormon women in even more passive positions than the Donner mother. In reality, while the practice of polygamy contributed to an exceptionally patriarchal society, it also afforded Mormon women opportunities for religious and civic leadership that were unusual for the era. Because many large polygamous families were spread across multiple estates, wives headed estates when their husbands were not present and collaborated to efficiently consolidate household work and childcare, freeing time for wives to also pursue education and establish careers (Beecher 1981, pp. 288–89). After abandoning the practice of polygamy near the turn of the twentieth century, however, Mormon women lost much of the institutional power that they had obtained in the nineteenth century, as the Correlation movement brought the formerly independent Mormon women's auxiliary, the Relief Society, under control of the all-male priesthood (Steenblik et al. 2016, pp. 24–25).

Despite this earlier abandonment of polygamy, in the 1940s, Latter-day Saints were still struggling to overcome the lasting public association of their religion with the practice.[9] Thus as he developed the This is the Place Monument in the 1940s, Mahonri Young was faced with the challenge of representing Utah's Mormon settlers as heroic, founding Americans—despite the fact that nineteenth-century Latter-day Saints maintained social practices that were widely excoriated by their contemporaries in the broader nation. While the Voortrekker frieze depicted African polygamy to reinforce racial divisions between white settlers and indigenes, Young carefully avoided the issue in This is the Place, giving no hint of the marriage practice that was already a defining feature of Mormonism by the time the settlers arrived. He accomplished this by minimizing the presence of women in the monument altogether—of the dozens of figures, there are only three named Mormon women and one anonymous female member of the Donner Party. The monument thus reverses the Mormon stereotype in American visual culture featuring Mormon men surrounded by throngs of women (Bunker and Bitton 1983, pp. 123–36).

Young seems to have justified the lack of Mormon women by focusing exclusively on the first company that arrived in the Salt Lake Valley on July 22–24, 1847, which was overwhelmingly male. The frieze on the front of the right central base depicts members of this company, most identified by name on the text panel below. The three female figures are named as Clarissa Decker Young, Ellen Sanders Kimball, and Harriet Page Wheeler Decker Young, who were the only three women in the 145-person company (Figure 10). Lest viewers imagine that the women were sister wives, the text also notes the names of each of the women's husbands, clarifying that Clarissa Decker and Harriet Page Wheeler Decker were married to two different Youngs (Clarissa to Brigham Young, and Harriet to his younger brother, Lorenzo Dow Young). In truth, all three women *were* polygamous wives—their sister wives came in later companies. Clarissa Decker Young and Ellen Sanders Kimball actually were members of the two largest polygamous families in early Mormonism, but the monument obscures this fact for viewers (Kimball 1986, p. 122; Johnson 1987, p. 60).

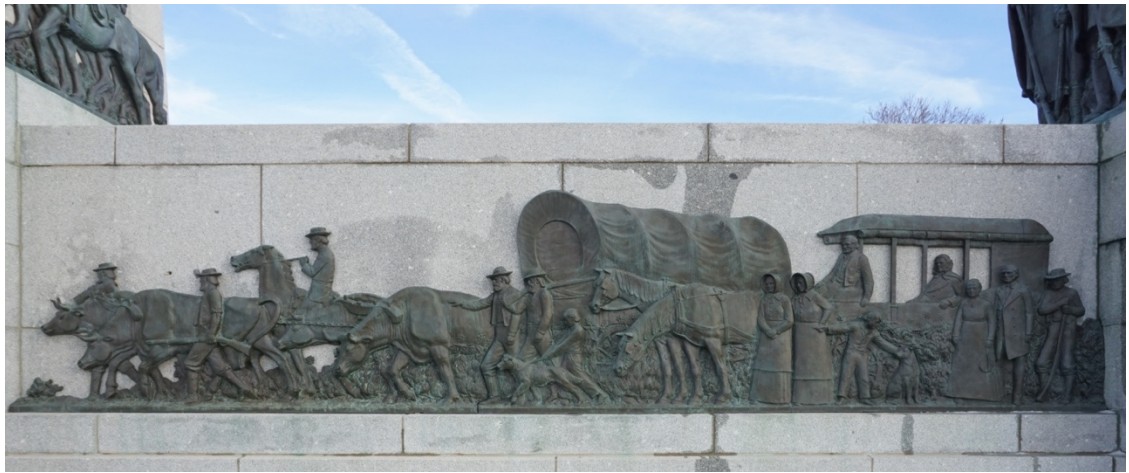

**Figure 10.** Mahonri Young, "First Pioneer Company" from This is the Place Monument, 1947. Photograph by Nathan Rees.

Just as This is the Place obscures the polygamous relationships of the women it represents, it is mute about the work that they did on behalf of Mormon migration and settlement, with the women only identified as wives. In fact, beyond the extraordinary work they undertook in raising families in a new colony where food and housing were dangerously inadequate, all three women were recognized by their community for their

---

9   For example, a popular 1941 Utah guidebook claimed that "More than any other feature of Mormon culture, polygamy has distinguished Utah and the Mormons in the public mind," noting that "Utahns are continually seized upon for information about polygamy." (Writers' Program of the Work Projects Administration 1941, p. 65).

heroic actions in supporting migration and settlement (Whitney 1904, vol. 4, pp. 63–69; Jakeman 1915, pp. 9–10). Unlike the three heroic men atop the central plinth, the three original female settlers are relegated to a servile role—despite their high class status, indicated by the carriage behind them—accepting the protection of their powerful male leaders. Juxtaposed with the Mormon men depicted ahead of them on the frieze along the monument's front, the women are strikingly passive, in what would have been seen as their natural role. While the men march forward, driving oxen, carrying rifles, or merely pointing forward, the women are rigidly stationary, their arms at their sides or quietly folded. Even the young boys are moving onward with their dogs—but the women just silently stand by.

As other white Americans embraced greater public roles for women, and even celebrated them in pioneer monuments, albeit in carefully conscribed ways, Mormon monuments placed women safely within nuclear families, presenting women who obediently follow the direction of active, prominent men. Minimalizing the presence and power of white women in the This is the Place Monument—and excluding indigenous women altogether—did more than just reaffirm colonization as a male-driven process—it helped reimagine Mormon settlers as core participants in the national project of nation-building, rather than radical others whose Americanness and even whiteness were suspect through their practice of polygamy.

## 4. Conclusions

Both the Voortrekker and This is the Place monuments employ raced and classed representations of hegemonic gender in order to support their cultures' superiority over indigenous and other white settler societies. Because gender was largely understood by mid-twentieth century audiences as natural rather than a cultural construction, monument builders promoted Mormon and Afrikaner male leaders and their male helpers as possessing a civilized manliness that justified their subordination of indigenes and superiority to other white settlers—both of which were shown as exhibiting deficient or improper masculinity. In their depictions of women, however, the monuments diverge as they each seek to valorize the communities they represent relative to the broader white societies of which they were a part. Each represent women in ways that serve that ideology, while excluding black and native women. Because both Mormons and Afrikaners had lost major conflicts with their white competitors in the late nineteenth century, both sought to rehabilitate their image in the early twentieth century through these monuments. However, where Afrikaners did so through celebrating the accomplishments of idealized white volksmoeders, Mormons erased the contributions of their female settlers in This is the Place in order to avoid referencing the polygamous social order that had simultaneously denigrated and empowered Mormon women and that also cast Latter-day Saints as radical exceptions to broader American gender ideals, both in the nineteenth century and in the twentieth.

Bringing the Voortrekker and This is the Place monuments into dialogue demonstrates the potential for the study of material culture to help elucidate the internal contradictions and struggles of communities as they define—and construct—race and gender through self-representation. As this work demonstrates, cross-cultural comparisons reveal the use of gender across settler societies as a tool in building racial identity and political ideology, since gender has been so widely understood as a natural order rather than a cultural construction. By interrogating the use of gender in constructing ideology through material culture, we can better understand the ideological work of settler monuments, while working to denaturalize gender and understand it as a social construction and a tool of hegemonic power.

**Author Contributions:** Conceptualization, investigation, and writing by C.P., N.R. and R.W.-H. All authors have read and agreed to the published version of the manuscript.

**Funding:** This research received no external funding.

**Conflicts of Interest:** The authors declare no conflict of interest.

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
