# Peer review of "Enshrining Gender in Monuments to Settler Whiteness: South Africa’s Voortrekker Monument and the United States’ This Is the Place Monument"

_humanities, doi:10.3390/h10010041_

Round 1

Reviewer 1 Report

This is a very interesting comparison that I do not think has previously been considered. I found it fascinating to learn of the simultaneous conception and creation in the 1930s and 40s of these monuments to commemorate the centenary of nineteenth-century pioneer achievements, and bolster the reputation of the pioneers they represent in the USA and South Africa respectively - all in the service of contemporary political agendas.

While I agree generally with the discussion of how masculinity is portrayed, I feel the differentiation of the Afrikaner leaders from other trekkers - as seems to be the case at the Utah monument - may be a little overstated. While it is true that three monumental trek leaders are portrayed on the exterior corners of the building, as is noted, an unknown Voortrekker of the same scale is also included. And there is less differentiation in the frieze itself, where all the Boers are invariably dressed with propriety, although leaders, especially Pretorius, may have more formal attire. (Is the difference of dress at Utah perhaps between the Donner and LDS menfolk?) However, it could be argued that the Voortrekker Monument with its cenotaph has a more 'sacred role', almost deifying the trekkers - it is not only a monument but a shrine to those who died.

As regards women, I find the argument that they have been treated differently at the two sites, despite their similar roles in their pioneering communities, particularly compelling. The different political agendas and the desire to underplay polygamy is a convincing explanation for their sparseness at Utah. As is pointed out, at the South African monument, on the other hand, women appear frequently and in a variety of roles, some of which may have been performed by black servants, who are suppressed in the frieze. This gives the women agency in the trek parallel to that of the men - perhaps even more so, since they bear the children that guarantee the succession of Afrikaners.

The indigenous figures that are frequently represented are not servants but savage enemies, another point of difference from the Utah monument. I would suggest that it might be more strongly acknowledged that the Voortrekker Monument has a  pronounced racist agenda - far more than This is the Place - presaging the apartheid legislation that the Afrikaner Nationalist government would introduce after its election in 1948. 

A few small points for noting:

Although not incorrect, it is rather misleading to refer to the VTM as concrete and steel, since this conjures up a modern edifice. It is only the sub-structure and the building presents a granite-faced exterior. And to say that it functions as a museum is not really correct – it is more of a shrine, although it has a museum area in the lower basement.

The Afrikaners won the first Anglo-Boer War although they lost the second.

Artist H.W. Coetzer did not really 'conceive' the frieze for the Voortrekker Monument: the architect introduced the concept, and Coetzer was invited to make initial sketches for it. The sculptors modified Coetzer's sketches considerably for the reliefs and added topics that he did not include.

I was curious to know whether bronze casting of high quality was readily available for the Utah monument. In the South African case, the bronze figure of the Voortrekker mother was the first large-scale casting in the country (by Italian immigrant foundrymen), and the plasters designed for the frieze were sent to Italy to be carved in marble.

Author Response

Response to Reader #1

We are pleased to hear that you find our comparison interesting and persuasive.  We would like to take the liberty to respond to the specific critiques that you offered.

We agree that the VTM almost deifies the trekkers, and it venerates them as a group more so than TITP, which venerates Brigham Young and other leaders more so than LDS settlers as a whole, or even the First Company featured on its front base.  We have clarified our analysis of the relative differentiation of Afrikaner and LDS leaders.  We find that Pretorius and other leaders appear in formal attire when paired with formally dressed Europeans, but in more informal jackets when contrasted against black African foes.  We have clarified this language throughout the essay.

We do not have space in this essay to adequately address the role of religion Voortrekkers and Mormon settlers.  However, we are also writing a companion article that focuses on intersections of race and religion in the same two monuments, so the religious meanings of each monument is something we have thought about a good deal.  This is the Place commemorates a religious group, but was erected and dedicated by a combination of church and state government officials.  One might argue that it seeks to venerate them also in a civic religion of American exceptionalism.  Meanwhile, the Voortrekker Monument is a secular-sponsored monument that operates as a shine, and is designed to celebrate what was known at the time of construction as the Day of the Vow.  

We sought to further emphasize the Voortrekker Monument’s racist agenda in our introductory framing of the essay.  We also added a brief discussion of the absence of indigenous women in This is the Place, which serves not only to clarify that point in regards to TITP, but also further emphasizes both our mention of VTM’s erasure of black women and its emphasis on white motherhood.

Thank you for bringing several inaccuracies in our essay.  We clarified the construction materials of VTM and of Coetzer’s role in designing the frieze.  We also clarified that the Afrikaners lost only the second Anglo-Boer war.

In regard to your query about bronze casting, it had been available in the United States since the nineteenth century.  Mahonri Young, like many other American sculptors used Roman Bronze Works, which was established in 1900 in a suburb of New York City. US commissions for marble carving, by contrast, were still often sent to Italy in this era.

Reviewer 2 Report

I have attached a pdf file with specific comments throughout the paper. 

Overall, I found this to be a very interesting and informative paper, demonstrating high level analysis of the the use of gender in creating racialized and classed interpretations in two cases of white settler history. I feel that the authors made a strong case for the methodologies employed by sculptors of the two monuments in directing altered views of history by manipulating cultural signals conveyed through monumentalization.

Two of my main suggestions (although both relatively minor) that I think would strengthen the piece both structural/organizational in nature. First, I would suggest highlighting some elements of the conclusion near the beginning of the paper, to frame the driving research questions inspiring the work. In the beginning of the paper, while the comparative structure and methodology of the work was made quite clear, I found myself wondering what (if any) the wider implications of this work might be, beyond the microanalysis of the two monuments themselves. A clear guiding problem or global question would help to better ground the authors' hypothesis and more succinctly line up the coming research trajectory.

Second, I think some reorganization of the inclusion of background summary might be helpful as well. Given the globally comparative nature of this work, I imagine many readers may be less familiar with one monument or the other, depending on their location and awareness of global history. Personally, as someone else familiar with the Boer wars and Afrikaner settlement, I found myself struggling in the first few pages to follow some indirect mentions of the historical conditions. I think that a more thorough summary of the historical situation of each monument near the beginning of the paper would increase overall clarity for the diversity of readership.

Finally, beyond structural issues, I was struck by the lack of attention given to the silence of indigenous women in the This is the Place monument. Considering substantial discussion was provided to the same near silence of representation of black women in the Voortrekker monument, the lack of discussion of this corollary in the Mormon monument created discrepancy in the parallelism of research, and seemed to be a substantial oversight. I would recommend at least adding an acknowledgement of the situation, if not a brief analysis of the implications of that omission in the sculpture as well.

These comments are suggestions only, and regardless I recommend the article for publication. I think it represents great and thorough scholarship, and makes substantial and important contributions to the way cultural material scholars consider the implications of gender and cultural manipulation in historically revisionist identity representation.

Author Response

Response to Reader #2

We are pleased to hear that you find our comparison interesting and informative.  We would like to take the liberty to respond to the specific critiques that you offered.

We have incorporated more of the concluding ideas into our statement of our thesis in the introduction.  We also added a bit more historical background to the introduction. These adjustments are admittedly minor due to space constraints, but we hope they will clarify our argument and historical positioning for the reader.

We appreciate you calling attention to our failure to discuss the silence of indigenous women.  We do not have space to explore this issue at length, but we did add an acknowledgment to that effect, which we believe substantially strengthens the essay as a whole.

We also greatly appreciate your in-line comments, and have done our best to respond to each one to the extent that space constraints permit.